Phylogenomics picks out the par excellence markers for species phylogeny in the genus Staphylococcus

Graña-Miraglia Lucia 1
Arreguín-Pérez César 2
López-Leal Gamaliel 3
Muñoz Alan 1
Pérez-Oseguera Angeles 1
Miranda-Miranda Estefan 2
Cossío-Bayúgar Raquel 2
Castillo-Ramírez Santiago iago@ccg.unam.mx 1
1 Programa de Genómica Evolutiva, Centro de Ciencias Genómicas, Universidad Nacional Autónoma de México , Cuernavaca , Morelos , Mexico
2 Centro Nacional de Investigación Disciplinaria en Parasitología Veterinaria del Instituto Nacional de Investigaciones Forestales, Agrícolas y Pecuarias , Jiutepec , Morelos , Mexico
3 Departamento de Microbiología Molecular, Instituto de Biotecnología, Universidad Nacional Autónoma de México , Cuernavaca , Morelos , Mexico
Hoyles Lesley
Electronic publication date: 2018 Oct 24
Publication date: 2018
Volume: 6
Electronic Location ID: e5839
Received 2018 Jun 4; Accepted 2018 Sep 28
Copyright: ©2018 Graña-Miraglia et al.
Copyright year: 2018
Copyright holder: Graña-Miraglia et al.
License: This is an open access article distributed under the terms of the Creative Commons Attribution License, which permits unrestricted use, distribution, reproduction and adaptation in any medium and for any purpose provided that it is properly attributed. For attribution, the original author(s), title, publication source (PeerJ) and either DOI or URL of the article must be cited.
License URL: https://creativecommons.org/licenses/by/4.0/

Keywords: Staphylococcus, Phylogenomics, Bacterial species, Infectious diseases, Evolutionary biology

Funding: CONACYT PDCPN 248049 INIFAP SIGI 1322633028 Programa de Apoyo a Proyectos de Investigación e Innovación Tecnológica PAPIIT IA201317 CONACYT 585414 CONACyT postdoctoral fellowship CB-2013/223279 This work was supported by CONACYT PDCPN (grant number 248049) and INIFAP SIGI (grant number 1322633028) to Raquel Cossío-Bayúgar and by “Programa de Apoyo a Proyectos de Investigación e Innovación Tecnológica PAPIIT” (grant number IA201317) to Santiago Castillo-Ramirez. The genome sequencing of the newly sequence isolates was conducted at Instituto Nacional de Medicina Genómica (http://www.inmegen.gob.mx). Lucía Graña-Miraglia is a doctoral student from the Programa de Doctorado en Ciencias Biomédicas, Universidad Nacional Autónoma de México (UNAM), and received a fellowship (number 585414) from CONACYT. Gamaliel López-Leal was supported by CONACyT postdoctoral fellowship CB-2013/223279. The funders had no role in study design, data collection and analysis, decision to publish, or preparation of the manuscript.

==============================
Although genome sequencing has become a very promising approach to conduct microbial taxonomy, few labs have the resources to afford this especially when dealing with data sets of hundreds to thousands of isolates. The goal of this study was to identify the most adequate loci for inferring the phylogeny of the species within the genus Staphylococcus; with the idea that those who cannot afford whole genome sequencing can use these loci to carry out species assignation confidently. We retrieved 177 orthologous groups (OGs) by using a genome-based phylogeny and an average nucleotide identity analysis. The top 26 OGs showed topologies similar to the species tree and the concatenation of them yielded a topology almost identical to that of the species tree. Furthermore, a phylogeny of just the top seven OGs could be used for species assignment. We sequenced four staphylococcus isolates to test the 26 OGs and found that these OGs were far superior to commonly used markers for this genus. On the whole, our procedure allowed identification of the most adequate markers for inferring the phylogeny within the genus Staphylococcus. We anticipate that this approach will be employed for the identification of the most suitable markers for other bacterial genera and can be very helpful to sort out poorly classified genera.

Introduction

The study of ecology and evolution has been substantially transformed by the omics technologies. Remarkably, the use of these technologies has allowed essential questions in evolutionary biology to be addressed and has advanced our knowledge in many biological processes (López-Leal et al., 2014; Joseph et al., 2015; Joseph et al., 2016). The precise identification of the different species within any given genus is highly valuable for many branches of microbiology. For instance, as far as clinical microbiology is concerned, this is instrumental in establishing which species are human/animal pathogens and which are just regular commensal organisms; whereas in microbial ecology species assignation is very helpful in defining the niche-range of the different species (Becker et al., 2016).

Although whole-genome sequencing is the best method for genotyping bacterial isolates and, therefore, to conduct species assignation/identification, this approach is not yet affordable for routine use in many laboratories all over the world. This is especially true for low and middle-income countries in which the amount of money invested in science is not as much as in developed countries. For this particular purpose, and from a theoretical point of view, orthologous genes are the perfect candidates for inferring the phylogeny of species, as they should reflect the species tree of the taxa considered. As per definition, orthologous genes are the ideal candidates to track the sequence of past of speciation events within a given lineage (Fitch, 2000). Therefore, in order to infer the phylogeny of the species, a clear-cut ascertainment of orthologous genes is of paramount importance and phylogenomic pipelines can be implemented to try to identify the potential orthologous genes. Orthologous relationships are context-sensitive and a gene family could be mainly composed of orthologous genes at one taxonomic level but if one considers a higher taxonomic level many more non-orthologous genes will appear. Furthermore, orthologous relationships could involve one to many relationships (Gabaldón & Koonin, 2013). The word co-orthologue was coined to describe that exact situation in which a genome has more than one orthologous gene (Gabaldón & Koonin, 2013). Hence, from a practical point of view (i.e., operational implementation), orthologous genes with just one gene per genome among the taxa considered should be the best markers to delineate the history of the species.

The genus Staphylococcus has a few dozens of species of Gram-positive bacteria, which are commensals colonizing the skin and mucous membranes of some mammals and birds. However, some of these species have a clear clinical and economic relevance, as they are a frequent cause of infection in humans, livestock, and domestic animals. Although the genus is very well known for the human opportunist pathogen S. aureus, which is famous worldwide as major source of nosocomial infections (Challagundla et al., 2018; Frisch et al., 2018), there are some other species such as S. epidermidis, S. lugdunesis, S. saprophyticus and S. schleiferi that have been associated with human infections. Several molecular-based methods have been introduced to carry out species identification within the genus Staphylococcus (Kwok & Chow, 2003; Ghebremedhin et al., 2008; Sasaki et al., 2010; Lamers et al., 2012). PCR along with sequence analysis of the genes dnaJ, tuf, sodA, rpoB, hsp60 and nuc have been used to differentiate Staphylococcus species (Kwok & Chow, 2003; Ghebremedhin et al., 2008; Sasaki et al., 2010; Lamers et al., 2012). Although they have been very useful and demonstrate a better resolution than the 16S rRNA gene (Ghebremedhin et al., 2008), these genes show different amounts of genetic diversity and, therefore, varying levels of discriminatory power for the different species. Thus, depending on the gene and the species, low-level resolution or even mis-identification can occur. Clearly, an ideal solution would be to conduct whole-genome sequencing of all the isolates considered to tell apart the different species. In terms of genome sequences, Staphylococcus is one of the few bacterial genera for which many genomes are available.

In this study, we address the question of which genes are the best inter-species markers (bona fide orthologous genes) for the genus Staphylococcus and, by applying a phylogenomic approach, we provide a list of the top candidates for species assignation within this genus. Underfunded groups could use these top candidates to accurately conduct species assignation without the necessity of using whole-genome sequencing. Furthermore, this approach could be used for many other species/genera to identify the most suitable markers for inferring the phylogeny of the taxa under investigation.

Methods

Genomes and homologous groups

We downloaded 265 publically available complete genomes (see Table S1). These cover 46 different species from the genus Staphylococcus and are a good representation of the host range within this genus. We ran CheckM (Parks et al., 2015) on these genomes to discard poorly sequenced genomes (i.e., with contamination or incomplete) and only one of them (marked in red in Table S1) did not pass the criteria (≥95% complete and with ≤5% contamination) and thus was not included in the rest of the analyses. We also sequenced the genomes of four bacterial isolates (see Table S2) obtained from hemolymph or hypostome exudates from adult ticks showing signs of bacterial infection and reared over experimentally infested bovines in the Centro Nacional de Investigación Disciplinaria en Parasitología Veterinaria (CENID-PAVET-INIFAP), Jiutepec, Morelos, México. Of note the four isolates sequenced were confirmed to be Staphylococcus spp by biochemical tests and 16S rRNA gene sequence analysis. The isolate INIFAP 005-08 was initially identified as S. saprophyticus by its physical and biochemical characteristics (Gram-positive coccus bacteria, catalase positive; novobiocin resistant, absence of coagulase, gelatinase and caseinase activity). This isolate was also able to produce acids by fermenting glycerol, lactose, D(+) mannose, sucrose and turanose; however it was unable to use L(+) arabinose or L(+) lactose. This isolate was classified as S. xylosus as per 16S rRNA gene analysis. The isolates INIFAP 002-16 and INIFAP 004-15 were identified as Gram-positive coccus bacteria, catalase positive and ribotyped by the 16S rRNA gene, which exhibited positive identity for S. xylosus. INIFAP 009-16 was identified as Gram-positive coccus bacteria, catalase-positive and identified as S. succinus using the 16S rRNA gene.  This same isolate was identified as S. carnosus using the API20E (bioMérieux  2010) system. The Nextera XT DNA Library Prep Kit was used for the sequencing libraries and Agilent High Sensitivity DNA Kit was employed for quality control. The isolates were sequenced using an Illumina MiSeq platform, with a 2 × 250 bp configuration; the genome sequencing was conducted at Instituto Nacional de Medicina Genómica (http://www.inmegen.gob.mx/) in Mexico City. Prior to assembling the genomes, we employed the SolexQA v.3.7.1 program (Cox, Peterson & Biggs, 2010) to trim the reads, employing the default parameters. We used Velvet version 1.2.09 (Zerbino & Birney, 2008) and Spades v3.11.0 (Bankevich et al., 2012) to carry out de novo assembly setting the option careful and we tested the following k-mer sizes: 51,61,71,81,91,101,111,121,127. We did not consider contigs smaller than 300 bp in the final assemblies. In all cases, Spades outperformed Velvet in terms of the number of contigs and the N50 statistics. Again, we used CheckM (Parks et al., 2015) to evaluate the quality of the genomes sequenced by us; notably all the 4 isolates were shown to be ≥95% complete and with ≤5% contamination and therefore reliable for downstream analyses. The details of the genomes assemblies are provided in Table S2. These seem to be good genome assemblies as judged by the high median coverage and the low number of contigs for each one of the newly sequenced isolates. These four genomes have been submitted to the GenBank and have the following accession numbers: PIZQ01000000, PIZN00000000, PIZP00000000, PIZO00000000 (BioProject PRJNA421192). Then, we employed PROKKA v1.11 (we set the genus option to Staphylococcus) to annotate all the genomes sequences (newly sequenced ones and the publically available) to have a consistent annotation for all of them. To get the orthologous groups (OGs) with one-to-one relationships among the species, we only considered single gene families (SGF); these were constructed running BLASTP searches with an e-value of 1.0 e−30 between the genome of S. aureus TW20 and the rest of the genomes. We kept all the cases where there was only one hit per genome and requiring that the seed from TW20 and the hit aligned ≥60% of their lengths and were ≥45% identical—the rationale behind these two criteria was to make sure that we had whole genes and not just domains. Then, for the all the SGF we constructed DNA alignments in frame via the program Fast Statistical Alignment version 1.5.9 (Bradley et al., 2009), setting the option –nucprot that align nucleotide sequences taking into account the protein space. We conducted recombination analysis on each of the SGFs using PhiTest (Bruen, Philippe & Bryant, 2006) that was implemented via the PhiPack program, using a window size of 50 bp. Additionally, we also determined the nucleotide diversity for each one of the SGFs using R pegas library function nuc.div() with the default parameters. We carried out a function enrichment analysis as follows: first, we conducted a Gene Onthology (GO) annotation via InterProScan version 5 using the genes from the S. aureus TW20 genome as a reference. Then, we conducted a GO enrichment analysis via Blast2GO PRO using a Fisher’s Exact Test (https://www.blast2go.com/), the analysis was conducted at three different GO levels: Molecular Function (MF), Biological process (BP) and Cellular Component (CC). Then to evaluate the results, we used a False Discovery Rate of 0.05 and the Benjamini–Hochberg correction was used to account for multiple testing.

Phylogenetic reconstructions, average nucleotide identity analysis and neighbour nets

We performed phylogenetic reconstructions for every SGF that did not show recombination signals as per PhiTest (Bruen, Philippe & Bryant, 2006) and for the super alignment (read below). All the gene trees were constructed with RaxML version 8.2.11 (Stamatakis, 2014) executing 10 inferences on the alignment, using 10 distinct randomized Maximum Parsimony (MP) trees and with the GTR+G+I model. Both the Shimodaira-Hasegawa topology test and the Robinson-Foulds distance were also conducted via RAxML with the default settings. It is known that species tree estimation is not a trivial matter and we employed a previous approach that gave trustable results (Castillo-Ramirez & Gonzalez, 2008). We created a super alignment concatenating all the SGFs that did not have signals for recombination and on this alignment a ML phylogeny was constructed also through RAxML, this time executing 20 independent inferences starting from 20 different MP trees and with GTR+G+I model. For this ML phylogeny bootstrap replicates were generated again employing RaxML, using the -x option that implements a fast algorithm for bootstrapping. The average nucleotide identity analysis (ANI) was run via the python module pyani (http://widdowquinn.github.io/pyani/), specifically we employed the ANIm method (Richter & Rosselló-Móra, 2009). To determine the GC content and the proportion of variable sites for each SGF, we used the function summary from the program AMAS (Borowiec, 2016). In order to visualize the conflicting phylogenetic signals in the genes rpoB and tuf we used SplitTree4 (Huson & Bryant, 2005) to construct Neighbour nets with uncorrected P distances.

Results

Defining the species tree, confirming genome affiliations and the set of orthologous genes

We focused on the genus Staphylococcus as it has clear clinical and veterinary relevance and, due to that, it has been extensively covered in terms of genome sequences. The data set employed for this study represents 46 species, incorporating 40 type strains, and included a total of 269 genomes for the analyses (see Tables S1 and S2). Notably, this data set has species with a wide host-range covering many of the niches described for this genus. First, we determined the single gene families (SGFs), as these are potential candidates to be orthologous genes with one-to-one relationships, and found 208 SGFs. However recombination could have affected them and, therefore, we conducted recombination tests on them and around 15% of them (31 SGFs) showed signals of recombination and were discarded, which left 177 SGFs as good candidates to be orthologous groups (OG); the list of these 177 SGFs is provided in Table S3. We employed a previously used strategy (Castillo-Ramirez & Gonzalez, 2008) to approximate the species tree. This is the total evidence approach, in which all the 177 SGFs without signals of recombination are concatenated and treated as if they were a single marker on which a Maximum Likelihood (ML) phylogeny was constructed (see Fig. 1)—this ML phylogeny was our Proxy for the Species Tree Topology (PSTT). From Fig. 1 one can see that type strains are scattered throughout the tree and that most of the isolates from individual species tend to form monophyletic groups and these groups are very well-supported as most of them have bootstrap values higher than 80 (see orange dots in the phylogeny). However, we also noted a region on the tree (shaded areas), where different species intermingle together. This is caused by two species, namely S. warneri and S. saccharolyticus, which do not form monophyletic groups and clearly some strains from these species have been mislabelled. Additionally, as an independent strategy, we also carried out an ANI analysis to calculate the relatedness of the strains to the type strains included in this study (see Fig. 2). This analysis shows that in terms of their taxonomy most of the genomes have been properly labelled. For instance, all the strains designated as S. lugdunensis clustered with the type strain, S. lugdunensis NCTC 12217, with identity percentages well above 95% and thus could be assigned to this species (Chun et al., 2018). The same applies to 78.26% of the other species; see for example S. gallinarum or S. simulans where genomes labelled as belonging to each of these species clustered tightly (again above the 95%) with the type strains. On the other hand, the mis-classification previously noted in the phylogeny is also evident in this analysis, as the strains of both S. warneri and S. saccharolyticus do not cluster all together in each case and their identity percentages are well below 95% (see red labels in Fig. 2). We then carried out a GO enrichment analysis to have an idea of the overrepresented functions of the 177 SGFs (see Table S4), as expected many of the enriched biological processes and molecular functions have to do with housekeeping functions; the top two GO molecular functions were ATP binding and GTP binding, whereas the top two biological processes were DNA repair and fatty acid biosynthesis (see Table S4 for more details). Taken together, these results demonstrate that the 177 SGFs seem to be real OGs and that most of the genomes have a proper affiliation regarding their taxonomy.

Figure 1 Maximum likelihood phylogenetic tree based on the super alignment of the orthologous groups.

Maximum likelihood phylogenetic tree based on the super alignment of the 177 non-recombinant SGFs. Strains of the same species are labelled with the same colour and black stars indicate type strain for some species. Green and violet rectangles highlight strains with conflicting clade assignment, S. warneri and S. saccharolyticus respectively. Bootstrap values above 80 are shown in orange dots. The scale bar shows the number substitutions per site. The code for the strain identifiers is in Table S1.

Ranking the orthologous groups

Then we analysed which of the 177 SGFs, from now on OGs, could be the best markers to infer the evolutionary relationships among the species. For that end, we established how similar each one of the single gene trees (from the 177 OGs) was compared to the PSTT; this was done employing the Robinson and Foulds (RF) distance between the single gene trees and the PSTT—this distance gives the number of bipartitions that are not shared by the two trees under consideration. We also used π, a common measure of genetic diversity, to establish the amount of genetic variation for each OG. Figure 3 gives the percentage of similarity of the 177 OG trees to the PSTT and the nucleotide diversity for each of the 177 OGs; this figure shows that no single OG yielded the same topology as the PSTT. Furthermore, we also noted that every single OG has its own topology—not shared by any other OG—as none of all the pairwise comparisons of the OG trees gave a RF equal to 0 (a RF distance of 0 implies that the two topologies in comparison are the same, see Fig. S2). Figure 3 also shows that 167 OGs (94%) have nucleotide diversity values higher than 0.15, which make them good candidates for phylogenetic markers. Of note, for two species (S. aureus and S. epidermidis) we also computed the intra-species diversity and it seems that these OGs even at this level have a good amount of genetic diversity (see Fig. S3). We found that the top 26 OGs were similar to the PSTT, as all of them have RF distances below 214, indicating that they are more than 60% identical to the PSTT. Notably, these 26 OGs present good values of nucleotide diversity (all but two showing values higher than 0.2 nucleotide changes per nucleotide site and the average for the 26 being 0.26 nucleotide changes per nucleotide site), which is very convenient for their use as to phylogenetic markers. Table 1 provides details about these OGs such as RF distance to the PSTT, nucleotide diversity, function and the ID in the S. aureus TW20 genome. Remarkably, when we concatenated these 26 OGs and constructed a phylogeny, this tree showed a topology pretty similar to the PSTT (see Fig. 4A), the RF distance was 74 being 86.1% identical to the PSTT. Furthermore, using only the best seven OGs (the bold candidates from Table 1), we got a similar result—that is the concatenated ML phylogeny of these seven OGs is almost 80% identical to the PSTT (see Fig. 4B). Here it is worth mentioning that although these two phylogenies (Fig. 4) are not 100% identical to the PSTT, the two phylogenies recovered almost all the species as monophyletic groups with very good bootstrap values (>80%), see blue dots in both phylogenies—the only exceptions being the miss-classifications noted also in the PSTT and ANI analysis. In considering this part, although no single gene tree of the 177 OGs yielded the PSTT, just using the best 26 (and even just the top seven) OGs we were able to recover a topology pretty similar to the PSTT.

Figure 2 Heat map of the Average Nucleotide Identity (ANI) analysis of the 269 strains from Staphylococcus spp.

The reddish cells show identity percentages above 95% implying that the strains belong to the same species, whereas non-reddish colors denote identity percentages below 95% (see ANI key, A). The rows on and by the heat map (B) show the species assignation (see Species key, C) and the dendograms show the clustering of the strains. Strain identifiers (D) are as in Fig. 1 and the color-coding is as follows: blue gives the type strains, whereas red shows the strains with issues of mis-classification. This analysis indicates 44 clearly discernible Staphylococcus spp.

Figure 3 Similarity to the species tree and nucleotide diversity.

Percentage of similarity between gene trees and the PSTT and the nucleotide diversity for each of the 177 SGF (green dots). The most similar gene tree topologies correspond to genes with high levels of nucleotide diversity. The commonly used marker genes dnaJ, rpoB and tuf are highlighted (orange dots), they all show low to moderate nucleotide diversity values and none of them show a similarity percentage to the PSTT above 60%.

Table 1 Top 26 genuine orthologous genes.

These are the most adequate markers for inferring the species phylogeny.

Descriptiona	NCBI reference sequence	GC content	Nucleotide diversity	Proportion of variable sites	Similarity with species tree (Robinson-fould distance)	
DNA mismatch repair protein (MutS)	WP_000073352.1	0.342	0.2370335	0.412	65.4135 (184)	
ATP-dependent RecD-like DNA helicase (recD2)	WP_001283311.1	0.345	0.2638535	0.588	65.4135 (184)	
Cytosol aminopeptidase (pepA)	WP_001009697.1	0.361	0.3404309	0.760	64.6617 (188)	
putative ABC transporter ATP-binding protein (YheS)	WP_000602071.1	0.339	0.2515735	0.688	64.6617 (188)	
Phenylalanine–tRNA ligase beta subunit (pheT)	WP_000908982.1	0.366	0.2683997	0.696	63.9098 (192)	
Penicillin-binding protein H (pbp3)	WP_000919772	0.340	0.2553319	0.623	63.9098 (192)	
DNA-directed RNA polymerase subunit beta’ (rpoC)	CBI48492.1	0.381	0.1488365	0.442	63.5338 (194)	
Pyruvate kinase (pyk)	WP_001232648.1	0.361	0.2139191	0.584	63.5338 (194)	
DNA polymerase III subunit alpha (DnaE)	WP_000226911.1	0.336	0.2895834	0.737	63.1579 (196)	
UDP-N-acetylmuramate–L-alanine ligase (murC)	WP_000150163.1	0.335	0.2352931	0.622	63.1579 (196)	
Fibronectin-binding domain-containing protein (YloA)	WP_000312763.1	0.338	0.2644582	0.667	62.782 (198)	
UDP-N-acetylmuramoyl-L-alanyl-D-glutamate-L-lysine ligase (murE)	WP_000340119.1	0.374	0.2366963	0.638	62.0301 (202)	
Homoserine dehydrogenase (dhoM)	WP_000735864.1	0.344	0.2720552	0.579	62.0301 (202)	
Acetyl-coenzyme A carboxylase carboxyl transferase subunit alpha (accA)	WP_000883645.1	0.357	0.2383029	0.584	62.0301 (202)	
Molybdopterin molybdenumtransferase (MoeA)	WP_000259718.1	0.386	0.2844009	0.679	62.0301 (202)	
S-adenosylmethionine:tRNA ribosyltransferase-isomerase (queA)	WP_001019171.1	0.357	0.2468871	0.620	61.6541 (204)	
ATP-dependent DNA helicase (RecQ)	WP_000983677.1	0.345	0.2654353	0.659	61.2782 (206)	
hypothetical protein (YibE/F-like)	WP_001794550.1	0.363	0.3498756	0.723	60.9023 (208)	
DNA polymerase III subunit tau (dnaX)	WP_001109047.1	0.368	0.2727606	0.645	60.9023 (208)	
Ribonuclease R (rnr)	WP_001050064.1	0.367	0.2284088	0.626	60.5263 (210)	
Ktr system potassium uptake protein B (ktrB)	WP_000021864.1	0.342	0.2674265	0.646	60.5263 (210)	
Dihydrolipoyl dehydrogenase (pdhD)	WP_001291535.1	0.353	0.3373007	0.759	60.5263 (210)	
3-oxoacyl-[acyl-carrier-protein] synthase 3 (fabH)	WP_001100524.1	0.377	0.2367141	0.601	60.1504 (212)	
Signal recognition particle receptor (FtsY)	WP_000007682.1	0.366	0.1979450	0.506	60.1504 (212)	
6-phosphogluconolactonase (hypothetical)	WP_000181322.1	0.358	0.3401093	0.737	60.1504 (212)	
Aspartate aminotransferase (AAT-like)	WP_001068542.1	0.342	0.2853785	0.678	60.1504 (212)	
Notes.

a Gene names were taken from the UniProt (http://www.uniprot.org) database when possible, otherwise the proteins were blasted against NCBI protein database (http://www.ncbi.nlm.nih.gov) and a well-annotated Staphylococcus species was used for annotation.

Figure 4 Phylogenies of the top orthologous groups.

Phylogenetic trees based on the concatenated alignments of the 26 orthologous groups listed in Table 1 (A) and on the concatenated alignments of the top seven orthologous groups (B). The tree in A has a percentage of similarity with the PSTT of 86% and the tree in B of 79%. Bootstrap values above 80 are shown in blue dots. Scale bar shows the number of substitutions per site.

The inadequacy of usual markers and the soundness of our strategy

Then, we wanted to know how a set of commonly used markers for inferring the phylogenetic relationships within this genus compared to our list. We chose the genes dnaJ, tuf, sodA, rpoB and hsp60 as these are the set of genes most used in previous studies (Kwok & Chow, 2003; Ghebremedhin et al., 2008). Most of them had clear issues to be considered suitable markers given the criteria we employed to define our genuine OGs. Notably two of them, tuf and rpoB, did have signals for recombination as shown by their Neighbour nets and the PhiTest (see Fig. S1) and their nucleotide diversity was very low (below 0.15, see Fig. 3 orange dots). On the other hand, sodA and hsp60 were not single gene families, as they did have more than one gene per genome in some species. Among these usual markers, only dnaJ seems to be a good candidate in as much as it is a SGF, did not have signals of recombination and has a nucleotide diversity value above 0.2 (see Fig. 3). The shortcoming of most of these genes is not totally unexpected, as these genes were not selected based on phylogenetic criteria; however, it is clear that most of these genes do not seem to be a good option to infer the history of the species.

Finally, to prove the validity of our phylogenomic approach, in the data set here employed we included the genome sequences of four bacterial isolates collected by us from cattle ticks and phenotypically classified as members of the genus Staphylococcus (see Table S2)—these isolates were sequenced using an Illumina MiSeq platform (see ‘Methods’). Initially these isolates were classified using the 16S rRNA gene sequence (see Table S2); however, in two of these cases, namely INIFAP 009-16 and INIFAP 002-15, the initial classification was incorrect. For instance, INIFAP 009-16 was classified as S. succinus by its 16S rRNA gene sequence but the PSTT placed this isolate with S. xylosus. In the case of INIFAP 002-15, whereas PSTT assigned it to S. succinus, the starting classification via the 16S rRNA gene was S. xylosus. Importantly, in the concatenate alignments using our top 26 OGs (Table 1) the isolates were properly classified (see Fig. 4A). Furthermore, the ML phylogeny based on the concatenated alignment of the top seven OGs also recovers the true affiliation of these isolates (see Fig. 4B). To sum up, our genuine OGs are a much better option than the usual markers to establish the phylogeny of the species and even just a few of these (the top seven) are able to adequately carry out taxonomy classification of the newly sequenced bacterial isolates.

Discussion

The main goal of our study was to establish the best markers for species identification in the genus Staphylococcus. To try to be as comprehensive as possible, we used a data set with 46 species and a total of 269 genomes from this genus that broadly represents most if not all the niches cover by this genus. Here we identified 177 OGs and ranked them according to their potential as phylogenetic markers; clearly, this set of markers should be useful not only for ecological and evolutionary studies but also for clinical and biotechnological purposes. In addition, our phylogenomic approach allowed us to identify two species (S. warneri and S. saccharolyticus) in which mislabelling of deposited genomes has occurred. This issue of misclassification is not that rare, as genome sequences submitted in public databases many times do not undergo a proper inspection in terms of microbial taxonomy. However, the combination of these phylogenomic approaches (genome-based phylogeny and ANI analysis) can be very useful to pinpoint those cases of misclassification.

We want to emphasise that our study has two major contributions: one is the list of adequate markers for inferring the phylogeny of the species within the genus Staphylococcus and the other is the phylogenomic strategy that we used to identify such markers. Regarding the first major contribution, given that the average gene content of the species here analysed is 2,413, our 177 OGs represent barely the 7% of the genome of these species. However, we need to emphasise here that, for merely practical reasons, we focused on orthologous genes with one-to-one relationships and, very likely, there are many more orthologous genes within these species that did not pass our criteria given that they are not SGFs. Despite the fact that all these OGs are potential good markers, in as much as their histories do not deviate significantly from the history of the species, they do differ among them in the amount of phylogenetic signal they each present. Importantly, no single OG was able to exactly represent the PSTT and no two OGs yielded the same topology; clearly, we found a cloud of topologies. This is in agreement with a previous study that found that only one of the several hundreds of OGs reflected the species tree (Castillo-Ramirez & Gonzalez, 2008), although this focused on different evolutionary scales and different bacteria. It is worth mentioning, that the top 26 OGs have good values of nucleotide diversity (all but two have values higher than 0.21), and are localized in different parts of the chromosome and present different functions. We think these 26 OGs represent a really good set of markers for inferring the PSTT; actually concatenating the best seven it was possible to get a very good approximation of the PSTT. Here, we want to highlight that using the phylogeny of the top 7 OGs all but the two species with issues of mis-classification were recovered as monophyletic groups and thus it seems that just using these seven markers species assignation can be carried out. Clearly, this set of markers could be very useful for underfunded laboratories working with Staphylococcus isolates. Although from a genotyping perspective, ideally one would want to use whole-genome sequencing for species identification, many laboratories in the world (specially in developing countries) still cannot afford the sequencing of tens to hundreds of isolates. Thus, this short list of markers should be extremely useful for those who only can sequence a few loci, which is often the case for clinical, environmental and evolutionary biology microbiologists in developing countries. Clearly, the markers highlighted in this study are a much better option than the commonly used markers, as the latter seem to exhibit obvious flaws (more than one copy per genome, signals of recombination, low nucleotide diversity values) for inferring the history of the species. Our strategy, and in turn the list of genes found, proved to be factually sound, even when using just a few markers for the taxonomic assignment of newly sequenced bacterial isolates.

Maybe more important than the first major contribution is the fact that our phylogenomic approach will allow the identification of adequate markers in many different genera—not only from bacteria but also from archaea. This is very important, as gene families showing only orthologous relationships very likely do not extend all over the tree of life—or just for a few gene families. Along these lines, it is very likely that many of the markers found here will not work for other genera (i.e., they will not show one-to-one orthologous relationships) as the homologous genes within those genera might have been affected by horizontal gene transfer or duplication and differential loss, or some other molecular event that prevent the history of the gene to reflect the speciation events. We want to highlight that the biology of the species under consideration could have a very important effect on the number of orthologous genes found. For instance, highly recombinogenic species, such as Neisseria gonorrhoeae (Ezewudo et al., 2015) and Acinetobacter baumannii (Grana-Miraglia et al., 2017), would have fewer orthologous genes than more clonal species. Nonetheless, the strategy employed by us should work even in these species, although the number of potential orthologous genes should be much less.

Conclusions

In summary, here we devised and applied a phylogenomic approach that allowed us to define the most suitable markers for inferring the species phylogeny within the genus Staphylococcus. We acknowledge that the effectiveness of these markers for other bacterial genera remains open to investigation. However, we are confident that the phylogenomic approach here implemented can be employed to identify the most suitable markers for other genera. On a broader level, this study has very practical implications, as it provides a general framework for mapping out the best markers to construct the phylogeny of the taxa considered not only at the genus level but also at other taxonomic levels.

Supplemental Information

Table S1 Publicaly available genomes used for this study

The 265 publicaly available genomes, from 46 species, were included, 40 of them are type strains (marked in bold). Genome statics were taken from NCBI. The third column gives the strain code used for figures 1, 2 and 4.The red bold row shows Staphylococcus saprophyticus UBA1057, which was removed from the analyses because the ChekM test indicated that its genome was not complete (completeness around 88%). The rest of the strains completeness is over 95% and contamination below 5%.

Click here for additional data file.

Table S2 Newly sequenced genomes for this study

Genome assembly statistics of thenewly sequenced Staphylococci strains.

Click here for additional data file.

Table S3 List of the 177 SGF without signals of recombination

Complete list of the 177 SGF without signals of recombination, which are our orthologous groups. Protein accession numbers were taken from the S. aureus reference genome TW20, the corresponding gene name was checked in Uniprot.

Click here for additional data file.

Table S4 Functional enrichment analysis

GO enrichment analysis via Blast2GO PRO.

Click here for additional data file.

Figure S1 Neighbour Nets and recombination test for rpoB and tuf

Phylogenetic networks built with SplitsTree, for two (rpoB and tuf) of the usual taxonomic markers. These markers are part of the 208 single copy genes but failed the recombination test. Strain labels are not shown for sake of clarity.

Click here for additional data file.

Figure S2 Robinson-Foulds distances between all the SGF trees

Histogram of the Robinson-Foulds distances estimated between all the SGF trees. There were no two identical topologies and on average topologies are 47% similar to each other.

Click here for additional data file.

Figure S3 Intra-species genetic diversity for S. aureus and S. epidermidis

Histograms of the intra-species nucleotide diversity for S. aureus and S. epidermidis.

Click here for additional data file.

We are extremely grateful to the editor and the reviewers, as their suggestions have drastically improved our manuscript.

Additional Information and Declarations

Competing Interests

Author Contributions

DNA Deposition

Data Availability

The authors declare there are no competing interests.

Lucia Graña-Miraglia performed the experiments, analyzed the data, contributed reagents/materials/analysis tools, prepared figures and/or tables, authored or reviewed drafts of the paper, approved the final draft.

César Arreguín-Pérez performed the experiments, contributed reagents/materials/analysis tools, approved the final draft.

Gamaliel López-Leal performed the experiments, analyzed the data, approved the final draft.

Alan Muñoz performed the experiments, approved the final draft.

Angeles Pérez-Oseguera performed the experiments, contributed reagents/materials/analysis tools, approved the final draft.

Estefan Miranda-Miranda and Raquel Cossío-Bayúgar contributed reagents/materials/analysis tools, approved the final draft, contributed bacterial isolates.

Santiago Castillo-Ramírez conceived and designed the experiments, performed the experiments, analyzed the data, contributed reagents/materials/analysis tools, prepared figures and/or tables, authored or reviewed drafts of the paper, approved the final draft.

The following information was supplied regarding the deposition of DNA sequences:

The 4 newly sequenced genomes have been submitted to the GenBank and have the following accession numbers: PIZQ01000000, PIZN00000000, PIZP00000000, PIZO00000000 (BioProject PRJNA421192).

The following information was supplied regarding data availability:

The four newly sequenced genomes are available under BioProject PRJNA421192.

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
