# Peer review of "Phylogenomics picks out the par excellence markers for species phylogeny in the genus Staphylococcus"

_PeerJ, doi:10.7717/peerj.5839_

## Round 0.1 · original submission · Major Revisions

Your manuscript has been reviewed by three experts. I agree with their comments, and provide my own below. Please address all comments when revising your manuscript. Your manuscript will be sent out for review again should you choose to submit a revised version to PeerJ.

The Abstract doesn’t really convey the aims of the study, and should be rewritten accordingly.

Introduction should include information on molecular-based methods (e.g. PCR, MLST) currently used to differentiate Staphylococcus species. Are any of the available methods able to be used across species, or are they specific to a single species? What are the limitations of currently available methods for typing members of the genus Staphylococcus.

You state in the Introduction there are many publically available genome sequences available for the genus Staphylococcus (>200 last time I looked in NCBI genomes). Therefore, why were only 64 sequences included in this study? All available type strain genome sequences for the genus Staphylococcus should be included in the analyses, and should be clearly marked as such in tables/figures. Refer to http://www.bacterio.net/staphylococcus.html if you are unsure as to which strains represent type strains for the genus.

To facilitate reproducible research, please address the following.
- provide details of the Illumina chemistry used, and the kit used (e.g. TruSeq, TruSeq-PCR, Nextera, etc.) to prepare the sequencing library.
- provide basic parameter information for SolexQA (including version), SPAdes (including version), Velvet and Prokka v1.11. MUSCLE version, TRANALIGN version.
- What was the rationale for choosing >=60 % coverage and >= 40 % identical for homologous groups? What other options were investigated? Report information in the manuscript.
- You report you used a False Discovery Rate of P < 0.05 for GO analyses. Please provide details of which method of FDR was implemented.

Use CheckM or Phylosift to generate completeness information for the new genome sequences you generated, and report these results in the manuscript (see https://www.nature.com/articles/nbt.3893 for other appropriate tools, if you are unable to use either of the aforementioned tools). Please refer to Yilmaz et al. (Nat Biotechnol. 2011 May;29(5):415-20. doi: 10.1038/nbt.1823), and confirm your data and their reporting comply with the minimum information requirements.

There is no information given as to how you confirmed affiliation of various genomes to the designated species. It is well known that sequences deposited in public databases have not necessarily undergone extensive checking (applies to NCBI and even RefSeq). Therefore, quality checks should be the first part of any pangenome analysis. At a minimum, average nucleotide identity (ANI) information is required (can be generated using fastANI or OrthoANI, with species delineation determined based on the recommendations of Chun et al. (2018; Int. J. Syst. Evol. Microbiol. 68, 461–466. https://doi.org/10.1099/ijsem.0.002516). The data can be presented as an ‘all against all’ heatmap, with information for genomes not affiliated with type strains or known authentic species strains based on ANI analysis reported in the text. Affiliation of genomes to species with no known type strain genome sequence should be confirmed by consultation to literature reporting specific genes that allow differentiation of species, and results reported in the manuscript.

Reviewer 1 ·

Basic reporting

Some minor spelling and grammatical issues, but English was clear and understandable. Raw data was shared.

Fig. 1 - do the authors have left and right the correct way around here? They say the ML phylogeny with the scale bar is on the right, but it is shown on the left. And that on the left hand side, numbers by the nodes reflect number of SGF trees having that partition, but the tree with the numbers by nodes is on the right. Also, what are the coloured branches supposed to signify?

Fig. 2 - the authors refer to RB distance in the legend, rather than R-F.

Supplementary table 3 would benefit from gene names (or TW locus numbers) for readers trying to find an individual gene in that table.

Supplementary table 4 lists FDR and P-values - it would be useful if the authors stated how many homology groups belonged to each GO.

Experimental design

The research question was well defined. The phylogeny provides an interesting update on previous studies, given the expansion of the number of described staphylococcal species in recent years.

However, the authors fail to justify their exclusion of a large number of orthologous genes from inclusion in the dataset used to generate the final concatenated "species" super-alignment phylogeny, that individual genes are tested against. Just because the individual genes' phylogenies differ significantly from the "species" phylogeny should not be sufficient on its own. Genes may evolve at different rates - if they are orthologous and single copy with no evidence of recombination, they shouldn't be excluded just because their phylogenies disagree from the species phylogeny. It's fine not to select them when considering individual genes to use as markers, but they should be included in the species phylogeny. On lines 164-165, the authors state "it is remarkable that both the super alignment and the consensus tree produced the very same topology" - but is it really remarkable, if you've removed all the genes whose phylogenies disagree with the super alignment phylogeny?

Validity of the findings

Lines 169-171: The authors state "as expected many of the enriched biological processes and molecular functions have to do with housekeeping functions..... Taken together, these results demonstrate that the 275 seem to be real orthologous groups". What gene ontologies did the homology groups that were discarded (because their phylogenies differed significantly) belong to? Do they differ to the "real" orthologous groups?

Lines 180-181: The authors state "every single orthologous group has its own topology - not shared by any other orthologous group", but there didn't seem to be any data presented to support this.

The authors also seem to contradict themselves in places, depending on the argument being made at the time. Line 209, they state "these two genes show a rather low genetic diversity". One of those genes, dnaJ, was amongst their top 17 markers. Lines 185-186, the authors had stated: "these 17 orthologous groups present very decent values of nucleotide diversity" and line 253, again seemingly referring to the same 17 orthologous groups: "these few genes have good values of nucleotide diversity" (although it's not entirely obvious here whether they're referring to the 17 genes, or the smaller subset of 4 genes, as they just refer to "these few genes". However, in the next sentence, they do refer to 17).

Lines 218-219: when discussing the four staphylococcal strains that were sequenced as part of the study, the authors state "in 2 of these cases, namely INIFAP 009-16 and INIFAP 002-15, the initial classification was incorrect". What is this based on? Are the authors just assuming that their phylogeny is correct, rather than the 16S RNA classification? Or have they performed any biochemical testing to determine the species type independently (also, how had the authors initially determined that these isolates were staphylococci, in order to know to sequence them and include in this study?)?

Lines 256-257: the authors state "furthermore, the particular combination of just three of them gave the exact same topology as the PSTT", but do not state which three genes they are referring to.

The maximum-likelihood phylogenies would benefit from bootstrapping, especially if one of the criteria being used is nodes shared by two trees. If these nodes are not supported, should they be perhaps collapsed? The authors could have used a number of other metrics in addition to just RF distance and nucleotide diversity. How many of the sites were informative? How many homoplasies were present for each gene? Which measure of nucleotide diversity was used, and have they used a correction (and did they use the DNAbin or haplotype options from the nuc.div function?)

In the discussion (lines 282-283), the authors state that "previous studies have shown that S. aureus and other species from this genus have not endure that many recombination events in the core genome". However, both references refer only to S. aureus. References for the "other species" would be beneficial given that some coagulase-negative staphylococci are thought to be quite recombinant.

Reviewer 2 ·

Basic reporting

Overall basic reporting is satisfactory.

However, the manuscripts need correction in english wherever required for example -

line no 58 -
within a giving lineage - within a given lineage (given to giving)

line 273

it is very likely that many of the markers here found - it is very likely that many of the
markers found here.. (found here to here found)

line 276 -

some other evolutionary that prevent the history of the gene to reflect
the speciation events. - some other evolutionary....? - incomplete sentence?!


number of orthologous genes there found. For instance, very recombinogenic species,
very - highly...
there found?

Experimental design

Overall the experimental design is satisfactory
Please see General comments to author

Validity of the findings

No comment.
Please see general comments to author

Additional comments

The authors propose a novel approach to identify powerful markers to infer robust phylogenetic relationship of bacterial strains from a particular genera. This is a major work in the field of phylogenomics and will be a reference for similar work other bacteria of importance in basic and applied research. The finding have not only implications in taxonomic or inter-species research but also in multilocus sequence typing and analysis studies.

However below are specific comments that need to be addressed before any decision is taken

1) Why no reference or type strains of species taken in to consideration?

2) Why no correlation with phylogenomic criterion for groups like Average Nucleotide Identity (ANI)?

3) What is the GC content of SFG and 17 markers genes? sign for HGT

4) There may be miss-classification in genera that are closely related to Staphyloccoccus. Why authors have considered using genera that are closely related to Staphylococcus...or can these makers help in identifying miss-classified and correct genus status of strains that are outlier or closely related to genus Staphylococcus?

5) How is the performance of 4 or 17 marker genes in case of MLST of important species like S. auerus and S. epidermidis? If not 5, query 5 is related and important and need to be a part of this present study.

5) Splits tree of the marker genes identified in the present study is not presented or discussed.

·

Basic reporting

The wording in the paper seems casual in many places and could be more concise.
The structure of the paper is fine however there are sections in the results that would be better suited for the discussion.
Species need to be in italic both in the actual paper, tables and references, as do gene names.
Scale bars on trees should be bigger, see Figure 1 left as an example.

Experimental design

The research question is interesting, however, the paper struggles between its two aims, none of which are clearly outlined in the abstract itself. Was S. aureus chosen because of the number of genomes available or because of its clinical significance?
The idea of determining representative genes to illustrate true phylogeny is interesting however their second aim was to present methods for non-WGS laboratories for proper species determination. Whereas the paper properly identifies such targets, it would have been suitable to define how to amplify these by PCR. See line 217-222 as an example where WGS should be applied despite this paper to obtain proper species assignment,
Figure 1 represents the alignment on the 275 SGFs, however, no information is presented on the pairwise distance of these, and additionally what is the diversity of these genes within each species? As mentioned, there are a vast amount of genomes available however these are not used to understand within-species diversity which could be important depending on inter-species distances. also, in refseq, I believe genome data of close to 50 species of Staphylococci are now available.
4 isolates are mentioned in the method section, but not why these were included? For validation?
Line 128: from what are the SGFs significantly different and what is the true tree to compare to?
Line 155-160: were the 663 trees manually compared? And what is the true history of the species for comparison?
Line 185: very decent value?
Line 237: phylogeographic studies?
Line 257: which 3?

Validity of the findings

The aims are too wide in my opinion - what is the true species tree for comparison, why is it useful to know this from a few genes from a clinical perspective. And the markers then decided to represent that is presented as a marker for non-WGS labs, but no methods like PCR primers are presented to allow that.
The paper needs to be more concise in language and larger sections in the Result part could be moved or incorporated into the discussion.
As for the markers, it would be interesting to get a basic knowledge of their intra-species diversity among the many available draft genomes of staphylococci to emphasize their stability (recombination) and lack of overlap between species.

Additional comments

See above.

---

## Round 0.2 · Minor Revisions

Thank you for taking on board my comments and those of the reviewers. Your manuscript has been improved significantly. There are still a few minor issues with the manuscript that need to be dealt with before it can be accepted for publication. Please refer to the reviewer's comments and also to the annotated PDF I have provided. Please ensure all cited literature is included in the reference list, and that formatting of citations is consistent throughout the manuscript. I look forward to receiving your revised manuscript in due course.

Reviewer 1 ·

Basic reporting

Some minor spelling and grammatical issues, but English was clear and understandable. Raw data was shared.

Good use of literature, but the authors do seem to switch between referencing styles on occasion (e.g. lines 187 and 249).

Experimental design

The authors have significantly strengthened the manuscript in this version, both in terms of the number of genomes analysed, and through the changes to their methodology.

Validity of the findings

No comment.

Additional comments

Minor Comments:
1. Would lines 111-119 (the biochemical results) be more appropriate in the results section?
2. Lines 124-5:"to trim the reads". What parameters were used here? What Q value? Reads greater than or equal to what length were kept?
3. Line 201: rather than 'the same applies to most of the other species', I think readers would appreciate a specific value here, e.g. 'the same applies to X% of the other species'
4. Line 205: 'see green labels in Figure 2'. There were lots of different green shades in this figure - the authors should find another way of referring to the relevant species in this figure.
5. Line 225-226: as for point 3, the authors should be more specific here. Rather than 'This figure also shows that most of the OGs...', use 'This figure also shows that X% of the OGs...'
6. Lines 231-232: include units for nucleotide diversity values 0.2 and 0.26.
7. Lines 236-237: 'the bold candidates from Table 1'. There are no bold candidates in table 1.
8. Line 265: 'the initial classification was incorrect' according to the PSTT. It would still be interesting to know what these species are assigned as by biochemical test.
9. Line 277: 'over 45 species'. The authors could just state the actual number here.
10. Line 304: 'almost exact same topology as the PSTT'. Are the authors over-stating this a little? Is 79% similarity (Figure 4) really almost exactly the same?
11. Figures 1 and 4 - the tree scales need units.
12. Figure 2 - why are there white boxes in the ANI for the unknown strains?
13. Figure 2 - would benefit from the phylogenetic tree on the left hand side of the figure also being labeled with colours for species - would make it much easier to interpret.

Spelling:
Line 78: Should be 'schleiferi', rather than 'scheiferi'.
Line 87: 'misidentification', rather than 'miss-identification'
Line 137: 'consistent', rather than 'consistency'
Line 169: 'python', rather than 'phyton'
Line 220: 'OG', rather then 'OGs'

---

## Round 0.3 · accepted · Accept

Thank you for dealing so promptly with the minor revisions requested by the reviewer. I look forward to seeing your article published soon.

#